


# A new scanning scheme and flexible retrieval for mean winds and gusts from Doppler lidar measurements

Steinheuer Julian[1,2], Detring Carola[3], Beyrich Frank[3], Löhnert Ulrich[1,2], Friederichs Petra[2,4], and Fiedler Stephanie[1,2]

[1]Institute for Geophysics and Meteorology, University of Cologne, Cologne, Germany
[2]Hans-Ertel Centre for Weather Research, Climate Monitoring and Diagnostics, Cologne/Bonn, Germany
[3]Deutscher Wetterdienst, Meteorological Observatory Lindenberg – Richard-Aßmann-Observatory, Lindenberg, Germany
[4]Institute of Geosciences, University of Bonn, Bonn, Germany

**Correspondence:** Julian Steinheuer (Julian.Steinheuer@uni-koeln.de)

**Abstract.** Doppler wind lidars (DWLs) have increasingly been used over the last decade to derive the mean wind in the atmospheric boundary layer. DWLs allow the determination of wind vector profiles with high vertical resolution and provide an alternative to classic meteorological tower observations. They also receive signals from higher altitudes than a tower and can be set up flexibly in any power-supplied location. In this work, we address the question of whether and how wind gusts

can be derived from DWL observations. The characterization of wind gusts is one central goal of the Field Experiment on Sub-Mesoscale Spatio-Temporal Variability in Lindenberg (FESSTVaL). Obtaining wind gusts from a DWL is not trivial because a monostatic DWL provides only a radial velocity per line-of-sight, i.e., only one component of a three-dimensional vector, and measurements in at least three linearly independent directions are required to derive the wind vector. Performing them sequentially limits the achievable time resolution, while wind gusts are short-lived phenomena. This study compares

different DWL configurations in terms of their potential to derive wind gusts. For this purpose, we develop a new wind retrieval method that is applicable to different scanning configurations and various time resolutions. We test eight configurations with StreamLine DWL systems from Halo Photonics and evaluate gust peaks and mean wind over 10 minutes at 90 m a.g.l. against a sonic anemometer at the meteorological tower in Falkenberg, Germany. The best performing configuration for retrieving wind gusts proves to be a fast continuous scanning mode (CSM) that completes a full observation cycle within 3.4 s. During this

time interval, about eleven radial Doppler velocities are measured, which are then used to retrieve single gusts. The fast CSM configuration was successfully operated over a three-month period in summer 2020. The CSM paired with our new retrieval technique provides gust peaks that compare well to classic sonic anemometer measurements from the meteorological tower.

## 1 Introduction

Extreme wind situations are an essential part of the weather-related hazards. The most important weather parameter to conclude

about associated damage is the wind gust peak (e.g. Pasztor et al., 2014; Jung et al., 2016; Schindler et al., 2016). Standard network observations and the parameterization in numerical weather prediction predominantly focus on gusts at 10 m height above ground (World Meteorological Organization, 2018; Brasseur, 2001; Schreur et al., 2008; Sheridan, 2011). A consequence





is that information on wind gusts from higher altitudes is rare. Though, the vertical resolution of wind gusts would help to better predict wind-related hazards and, in this context, identify vulnerable locations, which is useful, for example, for the design of

larger buildings or wind turbines.

The short-term nature of wind gusts makes it difficult to observe them accurately. According to the World Meteorological Organization (2018), a wind gust is a short-lived significant increase in wind speed that lasts for at least 3 s. A wind gust peak or briefly gust peak is the maximum wind gust in a certain time window, e.g. within ten minutes. A measuring device must therefore resolve the wind speed with high temporal resolution. The most advanced instruments are sonic anemometers,

which measure wind at sampling frequencies up to 100 Hz, although typical sampling rates for routine wind measurements at the National Meteorological Services are $1 - 4$ Hz. These are, however, in-situ measurements and the instruments have to be mounted to higher constructions. Further, the constructions influence the measurements, as can be observed, for instance, in the wake of wind turbines (González-Longatt et al., 2012). Long-term gust observations above 10 m are collected at meteorological tower sites that are equipped with sonic anemometers. The installation of a meteorological tower site is expensive and

regular maintenance is required thereafter. Mostly, this effort is made by research institutions and the national meteorological services at a few sites only e.g at Hamburg, Karlsruhe, or Cabauw (Brümmer et al., 2012; Kohler et al., 2017; Bosveld et al., 2020). Accordingly, the spatial coverage with such observations is sparse. Moreover, the height of such towers is limited to about 300 m, and hence no long-term observations could be made above this height.

Using Doppler wind lidars (DWLs) overcomes some of the limitations of meteorological towers. They are remote sensing

devices that have experienced growing interest as they have become less expensive in the last decades (Emeis et al., 2007). DWLs are portable instruments that can be set up with considerably less effort than for a tower. They provide reliable vertical profiles of mean wind in the lower atmospheric boundary layer under most conditions (Päschke et al., 2015). It is unclear, though, whether they are suitable for retrieving highly fluctuating gusts. A DWL measures Doppler velocities along different beam directions. To retrieve the three-dimensional wind vector, at least three measurements in linear independent directions

are needed. Thus, a DWL observes a wider volume of air to conclude on the wind vector. It follows from this fact that a DWL cannot provide a high-resolution time series of wind vectors at a given point in space as an in-situ instrument could. Very small-scale wind variations could be present only in certain of these air parcels. However, the gust peaks with the highest strength will also have a larger spatial extent and will influence different measurements and thus be more detectable. To obtain gust peaks comparable to the definition of a 3 s lasting wind peak and to the measurements of an sonic anemometer, a fast

measurement configuration for the DWL is required.

Suomi et al. (2017) propose a method for determining wind gusts using WindCube V2 DWL measurements. Their investigated DWL was scheduled for two days in a Doppler beam swinging (DBS) mode that provides measurements of five beams per one configuration cycle in 3.8 s. Wind gusts are derived from each five measurements, and gust peaks are obtained from them. The approach includes a scaling method to account for the lower temporal resolution than the 3 s definition. By this,

the results agree well with the 3 s-lasting gust peaks measured by nearby sonic anemometers on a meteorological tower. The considered observation period is very short and it remains open whether another measurement configuration is also suitable.





Within the Field Experiment on Sub-Mesoscale Spatio-Temporal Variability in Lindenberg (FESSTVaL, Fig. 1 (a)), different sub-mesoscale phenomena in the atmospheric boundary layer are investigated. These includes various observations and high-resolution modeling. Both address phenomena such as the evolution of the diurnal boundary layer, taking into account its

turbulent nature, and the evolution of wind gusts. For this purpose, multiple institutions gathered a variety of measuring devices in order to create a comprehensive observation network. Involved are a number of DWLs which were deployed at the boundary layer field site in Falkenberg next to a 99 m high meteorological tower. The tower is equipped with sonic anemometers which routinely provide wind and turbulence information.

In this study, we will focus on the deployed DWLs and their ability to retrieve wind gusts. Up to three colocated DWLs are

used to test different measuring configurations in parallel. The available DWL devices StreamLine from Halo Photonics cannot achieve the DBS scanning configuration in 3.8 s, but they are very flexible when it comes to setting up other measurement configurations. The results are compared with measurements from the sonic anemometer at 90.3 m. When exploring different DWL scan configurations, it turns out that a fast continuous scanning mode (CSM) is capable of completing a single revolution of the scanning head within 3.4 s. This configuration is thus closest to the gust definition and is therefore tested over an extended

period during the three summer months of 2020. For the calculation of the wind vectors, we develop a new retrieval scheme that can be used flexibly for different scanning configurations, for any number of observations, and for any desired time interval down to the duration of a single sampling cone. All results are derived from the new retrieval, and in addition to calculating the gust peaks, this is also used to determine the 10 min mean wind, since a practicable configuration must also correctly monitor the mean wind.

In section 2, we first provide an overview of the wind measuring devices, from which the data were obtained during FES-STVaL. Here we also describe in more detail the different tested DWL configurations. Section 3 introduces the new retrieval with an integrated iterative noise filtering. Section 4 provides the results and is structured in three parts that report on the test campaign in 2019/2020, that demonstrate the capabilities of the new retrieval scheme during the extra-tropical cyclone Sabine in February 2020, and that give a three months statistics on the DWL performance with the CSM in summer 2020. The paper

is concluded in section 5 and prospects plans for the evaluation of further FESSTVaL observations.

## 2   Wind measurements

The measurements analysed here are part of the FESSTVaL campaign. Originally, the FESSTVaL campaign was planned for 2020, but it had to be postponed to 2021 as a result of the Covid-19 pandemic, and its evaluation is not part of the presented work. Here, we will evaluate the 2019 test campaign and the reduced 2020 campaign, called FESST@MOL, in which fewer

measurements were made than initially planned but with DWL observations involved. In the 2019 test campaign, different configurations were investigated with up to three DWLs. In February 2020, the passage of extra-tropical cyclone Sabine could be observed with the same DWL which was operated for the following three summer months.

All instruments were operated at the boundary layer field site (in German: Grenzschichtmessfeld, GM) at Falkenberg (52°10′N, 14°07′E, 73 m above mean sea level). The GM Falkenberg is operated by the Meteorological Observatory



Lindenberg – Richard-Aßmann-Observatory (MOL-RAO) and is located about 5 km south of the Lindenberg observatory site, which is approximately 65 km southeast of Berlin, Germany (Fig.1, panel (a)). The measurement field is situated in flat terrain and is surrounded by agricultural land. There is a 99 m high meteorological tower at the field site, where sonic anemometers regularly measure wind and turbulence. Further information is given in subsection 2.1. The DWLs were deployed in about 70 m distance to the meteorological tower (Fig.1, panel (b)). Further information on the general measuring principle and the different configurations of the DWL are given in subsections 2.2 and 2.3.

### 2.1 Sonic wind anemometer measurements

The meteorological tower at the GM Falkenberg site is equipped with two sonic anemometers at 50.3 m and 90.3 m height. These ultrasonic wind anemometers are manufactured by Metek (factory version USA-1) and resolve the wind vector with a high temporal resolution of 20 Hz. Since the first usable DWL measurements are above 50.3 m, the measurement height of 90.3 m is taken as the reference for validation. To ensure data quality of the sonic anemometer measurements three main steps of operational data quality control are realized: filter nonphysical and constant values, detect spikes, and replace them by interpolating the neighboring points. The constant values can occur when the sonic anemometer is not working properly, for instance when it is frozen for a short time and sends the last measured value until a new measurement is available. Unrealistic spikes are detected following Vickers and Mahrt (1997) and replaced by a linear interpolation of the neighboring values. Despiking is very rarely used, and strong gusts are not removed by the procedure because they are characterized by a persistent signal in successive measurements that are technically no spikes. The filtered and corrected time series are used to calculate the 10 min mean and the 3 s lasting gust peak which is derived from a moving average over 60 single measurements within each 10-minutes interval. Thus the sonic anemometer gust peak represents a high-resolution reference for the DWL validation. The sonic anemometer at 90.3 m is located on a boom pointing towards South from the tower. The distance to the tower construction is 4 m. Due to shadowing effects caused by the meteorological tower itself, measured values from wind directions of $0° - 50°$ are disturbed and must be discarded in a fair evaluation. These are winds from the North-Northeast and thus from a not very frequently occurring direction in Falkenberg. For the comparisons of the sonic anemometer and the DWL measurements, only data from a wind direction sector between $50°$ and $360°$ is therefore analysed.

### 2.2 Doppler wind lidar measurements

A DWL measures radial wind velocities along the beam direction of emitted light in the near-infrared part of the electromagnetic spectrum. The emitted laser pulses are backscattered by aerosols and are received with a shifted frequency since the aerosols move with the wind. The range allocation of the backscattered signal follows from the travelling time. The Doppler shift in the light frequency enables the determination of the radial velocity, which is therefore referred to as the Doppler velocity. Figure 2 schematically illustrates the measurement principle. Each beam direction is determined by an elevation angle and an azimuth angle. The latter is counted clockwise from North, i.e. $0°$ equals North and $90°$ equals East. The beam is divided into a series of range gates. Each received Doppler velocity is assigned to the center of a range gate. The corresponding height of the center of the range gates depends on its distance to the sensor and on the inclination of the beam. To allow compari-



son with the sonic anemometer, we linearly interpolate for each beam a virtual Doppler velocity at 90.3 m from the retrieved Doppler velocities at the two nearest range gates. The wind retrieval presented in the following section is then also applied to
the interpolated Doppler velocities.

Three Halo Photonics DWLs have been part of the comparative test campaign - two of them (DWL 78 and DWL 177) owned by the German Weather Service (in German: Deutscher Wetterdienst, DWD) and one owned by the Technical University Berlin (DWL 143). A summary of their technical details is given in Table 1. The DWLs are flexible in setting up individual configurations. This involves the number of of pulses per ray, the number of radial measurements required by the DWL to
complete a single measurement cycle before repeating the configuration, as well as the elevation and azimuth of the beam direction. By using a smaller number of laser pulses per ray, a shorter duration to complete one measurement cycle is achievable. However, the accuracy of a single Doppler velocity may be reduced by using too few pulses. Typically, a DWL is operated in a step-stare mode, i.e. the DWL moves to an exact angular position, measures, and moves again, including acceleration and deceleration time. This time can be saved by setting up a continuous scanning mode where acceleration and deceleration are
omitted and measurements are taken during motion of the DWL scan head. Here, the azimuth covers a specific window, and each Doppler velocity is assigned to an azimuth representative of that window.

## 2.3 Doppler wind lidar configurations

We present eight different configurations that are tested for their suitability for retrieving gusts. Figure 3 illustrates the configurations with the corresponding panels as in the following itemized:

(a) CSM1 (75 s): Continuous scanning mode completing one DWL cycle in 75 s with 35.3° elevation angle. One measurement is performed with 3.000 pulses, and each cycle consists of about 210 beams, giving a relatively high spatial coverage. Smalikho and Banakh (2017) propose to measure continuously to determine the turbulent kinetic energy (TKE). The rather flat elevation angle of 35.3° is based on considerations by Eberhard et al. (1989), as this enables a convenient calculation of TKE.

(b) 24Beam (120 s): Step-stare mode in 120 s with 75° elevation angle. One measurement is performed with 30.000 pulses, and each cycle consists of 24 beams of exactly 15° azimuth steps to each other. This configuration is a popular mode for mean wind measurements with a relatively steep elevation angle to obtain observations from higher altitudes. At Lindenberg, for instance, there is another DWL that has been operated in this configuration for several years (Päschke et al., 2015). Similar to the CSM1, the 24Beam is not fast, but worth testing in terms of its widespread usage.

(c) DBS (28 s): Doppler beam swinging in 28 s with 62° elevation angle. One measurement is made with 30.000 pulses, and each cycle consists of four beams pointing North, East, South, and West, respectively, and one vertical beam. This configuration was proposed by Suomi et al. (2017) for measuring wind gusts, but originally with 3.8 s per cycle for the system used in their study. However, our Halo Photonics StreamLine DWLs do not reach this temporal resolution, but require 28 s to complete one cycle. Thus, although the study of Suomi et al. offers a promising way towards the retrieval





of wind gusts, it is not directly implementable here, and it is questionable whether we can achieve comparable results and hence validate their approach.

(d) 6Beam (35 s): Step-stare mode with six beams in 35 s with $45°$ elevation angle. One measurement is made with 20.000 pulses, and each cycle consists of five symmetrically arranged beams having an azimuth angle difference of $\Delta\theta = 72°$ with respect to each other, and one vertical beam. Sathe et al. (2015) propose to use this configuration for measuring
turbulence with a DWL. Six different measurements allow the estimation of the Reynolds stress tensor since it consists of six independent entries. Their approach concludes an optimal elevation angle of $\phi = 45°$ for the inclined beams.

(e) 3Beam1 (18 s): Step-stare mode with three beams in 18 s with $35.3°$ elevation angle. One measurement is made with 10.000 pulses, and each cycle consists of three beams having an azimuth angle difference of $\Delta\theta = 120°$ with respect to each other. A relatively short temporal resolution can be achieved by using only three beams for a DWL cy-
cle, and a relatively small number of pulses for a step-stare mode. Note that with using only three measurements, the calculation of the wind vector uncertainty is not possible and the result is prone to error and so a rather smaller elevation angle is chosen measuring the horizontal wind more directly.

(f) 3Beam2 (24 s): Step-stare mode with three beams in 24 s with $35.3°$ elevation angle. One measurement is made with 30.000 pulses, and each cycle consists of three beams having an azimuth angle difference of $\Delta\theta = 120°$ with respect to each other. Using only three beams but 30.000 pulses per beam gives this configuration duration of 24 s. It can
be seen that tripling the transmission rate does not increase the total cycle time that much, or vice versa, no time close to a 3 s-gust duration can be achieved with our devices in the step-stare modes.

(g) CSM2 (3.4 s): Continuous scanning mode in 3.4 s and with $62°$ elevation angle. The configuration uses 3000 pulses per measurement, which are assigned to an azimuth range and no longer directly to a defined constant beam direction. The
measurement is identified with a mean azimuth, and a complete cycle usually consists of 11 measurements, although due to the fact that the azimuth ranges drift, i.e. 10 or 12 counted measurements may also occur for some cycles. The high temporal resolution of 3.4 s is achieved when the beams are measured while the azimuth angle is continuously changing, and this mode of operation is clearly outperforming step-stare methods with respect to the cycle time.

(h) CSM3 (3.4 s): Continuous scanning mode in 3.4 s and with $35.3°$ elevation angle. One measurement is made with 3.000
pulses, and each cycle consists of roughly 11 beams. This fast continuous scanning mode uses a flat elevation angle of $35.3°$. The determination of an optimal elevation angle is not trivial. A higher elevation angle achieves larger measurement heights with smaller scanning cone cross section. With smaller elevation angle the horizontal wind can be measured more directly and the propagation of the measurement error can be reduced, but with the larger scanning cone cross section the assumption of wind field homogeneity can be violated already at smaller heights. This last configuration
is therefore in contrast to CSM2. The quick CSM can be challenging for DWL-hardware due to the rapid rotation.

The configurations were operated as illustrated in Table 2. The test campaign began in late summer 2019 and continued through autumn 2019. Extra-tropical cyclone Sabine in February 2020 is the most significant event in our observation period. Although





this event falls in 2020, it is likewise considered as part of the test campaign in the later analysis. The number of days shown does not exactly reflect the observation period, as the configurations were switched during the day and also some observations in the daily files were truncated at the beginning or end of the day. As the sonic anemometer does not provide valid observations for North-Northeast winds, these situations are missing in the comparison.

## 2.4 Noise filtering

Typically, a DWL wind retrieval begins with a preprocessing of the observations that filters out noise. There are several approaches that use the signal-to-noise ratio (SNR) to separate useful and noisy measurements (e.g Pearson et al., 2009; Barlow et al., 2011; Schween et al., 2014; Päschke et al., 2015). By comparing Doppler velocities with their SNR values, these approaches yield an SNR threshold at which measurements below this value should be discarded. The threshold is given at the highest SNR value where the Doppler velocities start to behave uniformly distributed over the entire range of theoretically measurable Doppler velocities, i.e. for our measurements roughly in the range of $[-20\,\mathrm{ms}^{-1}, 20\,\mathrm{ms}^{-1}]$ whereby $20\,\mathrm{ms}^{-1}$ denotes the approximate Nyquist velocity. This change in the distribution behaviour is most significant for direct measurements of vertical velocities, because they usually take values close to zero.

However, we have found that filtering by SNR threshold is not useful for some of our DWL configurations, especially for the quicker continuous scanning modes. Here, a high number of observations are achieved by emitting a relatively small number of pulses, which are then, however, associated with lower signal-to-noise ratios. If a threshold would be introduced and only the observations with SNR values below it can be assumed to be noise-free, many measurements would be discarded. Nevertheless, noisy values can also be observed for the CSMs over the entire SNR range, which is why a rigid threshold value does not seem appropriate for this reason either. And in addition, threshold filtering always has the problem that too many measurements with reasonable Doppler velocities are eliminated.

As an example, Fig. 4 illustrates all SNR values measured with CSM2 on September 2, 2019 against their Doppler velocities. Here, it should be noted that the zenith angle is $28°$ so that the vertical wind is not measured directly. One can assume, however, that the very high absolute Doppler velocities correspond to noise. In this case, it is appropriate to detect noise by absolute values that are above about $5\,\mathrm{ms}^{-1}$. The two vertical lines in Fig. 4 are examples where an SNR-threshold could be set, e.g at $-23\,\mathrm{dB}$ as done by Pearson et al. (2009) or at $-18.2\,\mathrm{dB}$ as done by Päschke et al. (2015). Nevertheless, at any reasonable or calculable threshold, noise would still be present in the measurements filtered this way, even if we would filter at an even stricter threshold, i.e., at a vertical line that would be further to the right in Fig. 4. And conversely, it can be seen that a large proportion of the measurements are in a region where the SNR thresholds suggest unreliable values (red region).

Instead of filtering the measurements in advance, we develop a method that initially includes all measurements, but then iteratively filters out those measurements that deviate significantly compared to an intermediate fit-solution such that they are detected as noise. This ensures that enough data are available to derive wind and, in particular, gusts, which are in fact based on very few measurements. Simultaneously, the iteration incorporates thresholds that terminate the retrieval procedure if the set of measurements is too inconsistent and conditions prevail under which the wind vector cannot be derived. The complete iteration procedure is explained in more detail in the next section, as it is integrated in the retrieval.





## 3 Retrieval

The following calculations can be made for measurements performed during a specific time window, such as a 10 minute interval, or based on measurements during a single DWL cycle. The number of single measurements per DWL cycle depends

on the used configuration.

### 3.1 Wind vector fit

A measured Doppler velocity $d_i$ is the projection of the wind vector $\mathbf{v}_i$ along the measuring beam direction $\mathbf{a}_i$ and satisfying the relation

$$d_i = \mathbf{a}_i^T \mathbf{v}_i + \epsilon_i, \tag{1}$$

with $\mathbf{a}_i = (\sin(\phi_i)\sin(\theta_i), \cos(\phi_i)\sin(\theta_i), \cos(\theta_i))^T$ where $\theta_i$ is the zenith and $\phi_i$ the azimuth angle of the $i$th of $i = 1 \ldots n$ consecutive Doppler velocity observations at a certain height. The instrument induced observation errors are $\epsilon_i$, which are assumed independent and normally distributed with zero mean and variance $\sigma_\epsilon^2$. The different Doppler velocities $d_i$ originate all from different beams and thus from different wind vectors $\mathbf{v}_i$. Since the measurements are made sequentially, with changing azimuth angle, there is not only a spatial but also a temporal difference, which is reflected in the $\mathbf{v}_i$. However, we assume that

the wind field is homogeneous and each $\mathbf{v}_i$ in the given time window, i.e. including the single DWL cycle, is the realization of one multivariate normally distributed random variable

$$\mathbf{v}_i \sim \mathcal{N}(\mathbf{v}_0, \Sigma), \tag{2}$$

with mean wind vector $\mathbf{v}_0$ and three-dimensional covariance matrix $\Sigma$. The homogeneity assumption may be violated over complex terrain or during long time intervals. The different $\mathbf{v}_i$ are assumed independent, which is another strong assumption

and should be scrutinized by a DWL user as it ignores spatial and temporal correlations.

With different realizations $\mathbf{v}_i$, i.e. with consecutive measurements at different viewing angles $\theta_i$ and $\phi_i$ in a certain time window, the underlying values $\mathbf{v}_0$ and $\Sigma$ could be estimated. The Doppler velocities $d_i$ then are the linearly transformed wind vectors (i.e. projection on beam direction in Eq. (1)), with an error variance that represents the observation error $\epsilon_i$ as well as the projected wind vector variability. They are normally distributed according to

$$d_i \sim \mathcal{N}(\mathbf{a}_i^T \mathbf{v}_0, \mathbf{a}_i^T \Sigma \mathbf{a}_i + \sigma_\epsilon^2). \tag{3}$$

We now assume that the wind vector variability is isotropic, i.e. the deviations of the individual $\mathbf{v}_i$ from $\mathbf{v}_0$ are identically distributed in all spatial directions. Then the projection of the covariance matrix is independent of the direction $\mathbf{a}_i$ and

$$\mathbf{a}_i^T \Sigma \mathbf{a}_i = \sigma_v^2. \tag{4}$$

The variance of $d_i$ is thus a combination of the measurement error and of the projected wind variability, i.e. the representation

error. The likelihood function $L$ for $i, \ldots, n$ measured Doppler velocities $d_i$ then reads

$$L(d_1, \ldots, d_n; \mathbf{v}_0, \sigma^2) = \prod_{i=1}^{n} f(d_i; \mathbf{a}_i^T \mathbf{v}_0, \sigma^2), \tag{5}$$





where $\sigma^2 = \sigma_v^2 + \sigma_\epsilon^2$ is the combined variance and $f(x; \mu, \sigma^2)$ is the probability density function of a Gaussian distribution with expectation $\mu$ and variance $\sigma^2$. Storing the $n$ different beam directions $\mathbf{a}_i$ row-wise in a $n \times 3$-matrix $A$ and the Doppler velocities in an $n$-dimensional vector $\mathbf{d}$ yields the maximum likelihood estimate (MLE) for $\hat{\mathbf{v}}_0$ which is

$$\hat{\mathbf{v}}_0 = (A^T A)^{-1} A^T \mathbf{d}, \quad \text{for} \quad n \geq 3. \tag{6}$$

Thus, $\hat{\mathbf{v}}_0$ is the least-squares fit over all measurements $n$ within one single DWL cycle or within a respective time window. Note that we need at least three independent beam directions for the inversion of $A^T A$. The residuals $e_i = d_i - \mathbf{a}^T \mathbf{v}_0$ can be used to estimate $\sigma^2$. For this, we use the unbiased estimator, i.e. the denominator $n - 3$ instead of $n$ to account for the degrees of freedom used to estimate the components of $\hat{\mathbf{v}}_0$, which leads to

$$\hat{\sigma}^2 = \frac{1}{n-3} \sum_{i=1}^{n} (d_i - \mathbf{a}_i^T \hat{\mathbf{v}}_0)^2, \quad \text{for} \quad n > 3. \tag{7}$$

In case of exactly three measurements the estimation of the variance $\hat{\sigma}^2$ is not possible. The corresponding standard deviation $\hat{\sigma}$ is equivalent to the root-mean-squared error (RMSE) and gives a measure of the fit performance.

## 3.2 Distribution of the estimator $\hat{\mathbf{v}}_0$

With all the assumptions, the residuals are Gaussian distributed with zero mean and variance $\sigma^2$. Latter results from the assumption that the variability of the wind vector $\mathbf{v}_i$ and the Gaussian observation errors $\epsilon_i$ are independent. Under the assumptions above, the distribution of the estimator $\hat{\mathbf{v}}_0$ is multivariate Gaussian distributed. The expected value of $\hat{\mathbf{v}}_0$ is given as

$$E[\hat{\mathbf{v}}_0] = \mathbf{v}_0. \tag{8}$$

The expectation value estimator is therefore unbiased. The variance of $\hat{\mathbf{v}}_0$ is

$$Cov[\hat{\mathbf{v}}_0] = (A^T A)^{-1} \sigma^2 \tag{9}$$

Both moments are in detail derived in the Appendix. Note that $A^T A = \sum_{i=1}^{n} \mathbf{a}_i \mathbf{a}_i^T$ and the number of rows increase proportionally with $n$. One important assumption behind the covariance estimate of $\hat{\mathbf{v}}_0$ is that $\mathbf{v}_i - \mathbf{v}_0$ are independent of each other, and the number of independent observations (i.e. degrees of freedom, DOF) is $n - 3$. This is definitely not the case, since the number of effective DOF $n_{ef}$ is much smaller than $n - 3$ and therefore $\sigma$ represents a lower bound of uncertainty. If we now assume that the estimate is based on substantially fewer independent measurements, we need to introduce a correction factor and estimate the covariance matrix $\hat{\Sigma}_{\hat{\mathbf{v}}_0}$ with an effective $n_{\text{ef}}$ instead of $n - 3$, reading

$$\hat{\Sigma}_{\hat{\mathbf{v}}_0} = \frac{n-3}{n_{ef}} (A^T A)^{-1} \hat{\sigma}^2. \tag{10}$$

Here, the estimate $\hat{\sigma}^2$ in Eq. (7) is used, and the $n_{ef}$ needs to be specified depending on the desired time window of the retrieval.



### 3.3 Iterative retrieval update

Our retrieval is aiming at the estimation of two variables $v_m$ and $v_g$. The 10 min mean wind velocity $v_m$ is estimated according
to Eq. (6) over all $n_{10}$ beams within a 10 min interval. The wind gust peak of a 10 min interval $v_g$ is the maximum of wind
estimates, each derived from measurements along a single DWL cycle with $n_c$ observations, again using equation (6).

As discussed before, the noise in DWL measurements is uniformly distributed over the measurable Doppler velocity range,
and therefore distort the estimation of $\hat{\mathbf{v}}_0$. This is the case when $\hat{\sigma}$ is particularly large. For example, pure noise with uniformly
distributed observations within $[-20\,\mathrm{ms}^{-1}, 20\,\mathrm{ms}^{-1}]$ would yield an estimate of $\hat{\sigma} \approx 11.6\,\mathrm{ms}^{-1}$. Our retrieval procedure aims
at filtering out the Doppler velocity measurements $d_i$ that are dominated by noise in an iterative process. To this end, we define
a threshold $u_1$ for $\hat{\sigma}$ at which the $\hat{\mathbf{v}}_0$ is assumed to be dominated by noise, as well as a minimum number $q$ of measurements $d_i$
that should be included in the estimation of $\hat{\mathbf{v}}_0$. If $\hat{\sigma} > u_1$, then $\hat{\mathbf{v}}_0$ is not accepted and the $r$ measurements with largest absolute
residuals $e_i$ are removed. Provided that the number of remaining $d_i$ is not less than $q$, $\hat{\mathbf{v}}_0$ is estimated again. Otherwise $\hat{\mathbf{v}}_0$
should be regarded as dominated by noise and set to not-available (n.a.). However, we introduce a second threshold $u_2$ which
is more tolerant and accept $\hat{\mathbf{v}}_0$ if $\hat{\sigma} \leq u_2$ even though $\hat{\sigma} > u_1$. This second threshold is a higher bound at which sufficient
confidence in the result has already been achieved, and the first threshold is a lower threshold that allows further improvement
of the estimate when enough data are available. Note that the parameters $u_1$, $u_2$, $r$, and $q$ are different for the two wind variable
estimates $v_m$ and $v_g$.

The iteration procedure is displayed in Fig. 5. In the upper right, the parameters are displayed for both $v_m$ and $v_g$. The
termination criterion $u_1$ is $\hat{\sigma} \leq 1\,\mathrm{ms}^{-1}$ in both cases. For $v_m$ the second threshold is $u_2 = 3\,\mathrm{ms}^{-1}$. Since the single-cycle
estimates of $\mathbf{v}_0$ relies only on very few $d_i$, we do not let $u_2$ be more tolerable, i.e. $u_2 = u_1 = 1\,\mathrm{ms}^{-1}$. We in turn require that
at least 66 % of the measurements are included for the single-cycle iteration, while $q = 50\,\%$ is sufficient for the 10 min mean
wind. The number $r$ of discarded measurements per iteration is 5 % for the 10 min wind and one for the single-cycle estimates.
The set thresholds are intended to provide a clear distinction between observations that are too noisy and those which are
usable. Nevertheless, it is possible to tune these values, but this is beyond the scope of this work.

Figure 6 illustrates the principle of the iteration procedure. Panels (a) to (c) illustrates different iteration steps for the es-
timation of a 10 min mean wind and panels (d) to (f) for the estimation of a wind of one single cycle. In the upper panels,
the iteration runs for ten complete iterations, discards 50 % of measurements, and ends with $\hat{\sigma}$ that falls in between $1\,\mathrm{ms}^{-1}$
and $3\,\mathrm{ms}^{-1}$. Hereby, panel (b) shows the first intermediate state where the retrieval would already accept the estimate because $\hat{\sigma}$
falls below $3\,\mathrm{ms}^{-1}$, but then continues improving until less than 50 % of measurements are used, as here in (c), or the more
rigorous threshold $u_1 = 1\,\mathrm{ms}^{-1}$ would be reached. In the lower panels, one measurement is discarded in each iteration and the
retrieval only returns a result that falls below $1\,\mathrm{ms}^{-1}$, as here in panel (f), since the $\hat{\sigma}$ in (d) and (e) are both too high.

Combined, the retrieval then provides the 10 min mean and a sequence of cycle-based individual winds within 10 minutes.
The gust peak can then be determined from the cycle-based winds. Here another check is included to prevent being susceptible
to unrealistic outliers. Those cycle-based winds that deviate in absolute speed by more than $1\,\mathrm{ms}^{-1}$ from all others within
the 10 min-sequence are removed. This affects outliers, of the two bounds, so both the strongest and weakest gusts are checked.





If at least 50 % of the cycle-based winds still exist and also the 10 min wind is not n.a., the gust peak is then determined to be the maximum of all remaining cycle-based winds within 10 min (and the minimum is defined as the minimum of the cycle based winds).

### 3.4 Estimation of uncertainty

The covariance estimate $\hat{\Sigma}_{\hat{\mathbf{v}}_0}$ in Eq. (10) includes the estimated value $\hat{\sigma}^2$. If $\hat{\sigma}^2$ is derived from the residuals that remain after the iteration process to estimate $\hat{\mathbf{v}}_0$, then the uncertainty is greatly underestimated. However, the inclusion of all measurements would overestimate the uncertainty. To account for uncertainty in the eliminated observations that is consistent with our statistical model, we assume that these residuals represent the truncated part of a normal distribution. Therefore, the variance $\hat{\sigma}^2$ estimated from the non-eliminated measurements must be corrected accordingly. Let $p$ be the percentage of discarded measurements, i.e. truncated values. If $p$ is the fraction of two-sided truncated values at symmetric thresholds $a$ and $b$, then the threshold $a$ and $b$ are given by $(a - \mu)/\sigma_S = \Phi^{-1}(p/2) = \alpha$ and $(b - \mu)/\sigma_S = \Phi^{-1}(1 - p/2) = \beta$, i.e. the $p/2$ and $1 - p/2$ quantiles, respectively, where $\Phi$ is the cumulative distribution function and $\phi$ the probability density function for the standard normal distribution of the original (non-truncated) values with parameters $\mu$ and $\sigma_S^2$. Following Johnson et al. (1994), the relation between the variance of the truncated variable $\sigma_T^2$ and non-truncated $\sigma_S^2$ is

$$\sigma_T^2 = \sigma_S^2 \left[ 1 + \frac{2\alpha\phi(\alpha)}{\Phi(\alpha) - \Phi(\beta)} \right] = \sigma_S^2 \left[ 1 + \frac{2\Phi^{-1}(p/2)\phi(\Phi^{-1}(p/2))}{1 - p} \right]. \tag{11}$$

This can be used to re-scale $\hat{\sigma}$ and approximate a corrected covariance matrix, towards

$$Cov\left[\hat{\mathbf{v}}_0\right] = \frac{n - 3}{n_{ef}}(A^T A)^{-1}\hat{\sigma}^2 \left[ 1 + \frac{2\Phi^{-1}(p/2)\phi(\Phi^{-1}(p/2))}{1 - p} \right]^{-1}. \tag{12}$$

We use the corrected covariance matrix as the estimate of the wind uncertainty for both the 10 min mean wind and the wind of a cycle. The uncertainty of the gust peak is associated with the covariance matrix of the corresponding maximum. Determining $n_{\text{ef}}$ is discussed in subsection 4.3.

### 4 Results

The results in subsection 4.1 are obtained from DWLs operated in different configurations from the end of summer 2019 to the beginning of winter 2019/2020, with additional consideration of three days of Cyclone Sabine in February 2020. Moreover, Cyclone Sabine is the subject of subsection 4.2. Based on these results, we performed measurements in the fast CSM2 over several weeks in summer 2020, for which performance statistics were derived in subsection 4.3.

### 4.1 Comparative test study

Figure 7 shows scatterplots of the 10 min mean horizontal wind from sonic anemometer versus the DWL retrieval for the eight configurations in Fig. 3. In order to measure the quality of the retrieval, we use the RMSE, the bias, and the coefficient of determination $R^2$ between DWL retrieval and sonic measurement. All eight configurations produce only minor biases, ranging from $-0.13\,\mathrm{ms}^{-1}$ to $0.14\,\mathrm{ms}^{-1}$. The CSM1 in panel (a) is based on a large sample since it has been tested almost all





the time. Apart from some underestimations at low wind speeds, here the wind is observed with small RMSE ($0.41\,\mathrm{ms^{-1}}$),
high $R^2$ (0.97), and negligible bias ($0.04\,\mathrm{ms^{-1}}$). For the 24beam in panel (b), some DWL outliers can be recognized, which
can be explained by the relatively steep elevation angle. The outliers result from the fact that the linear interpolation of the
Doppler velocities at 90.3 m fails because the involved Doppler velocities of the lower range gate centers are very close to
the DWL, where more unreliable measurements occur. Here the range gate centered in 75 m radial distance from the sensor
is involved, which has to be considered not always reliable. In fact, a comparison with the results of range gates centered
at 101.4 m would give a better result (not shown). These outliers lead to a higher RMSE ($1.12\,\mathrm{ms^{-1}}$) and a lower $R^2$ (0.8).
DBS in panel (c) and 6Beam in panel (d), both exhibit low RMSE values ($0.29\,\mathrm{ms^{-1}}$ and $0.34\,\mathrm{ms^{-1}}$, respectively) and $R^2$ val-
ues close to 1 (both 0.98), indicating a low scattering between sonic anemometer measurement and DWL retrieval. The 3Beam
configurations in panels (e) and (f) perform very similar and the scatterplots are based on parallel performed measure-
ments in October 2019. The quicker configuration actually performs slightly better in terms of diagnostic variables (RMSE
with $0.38\,\mathrm{ms^{-1}} < 0.48\,\mathrm{ms^{-1}}$ and bias with $|0.0\,\mathrm{ms^{-1}}| < |-0.11\,\mathrm{ms^{-1}}|$), although this is mainly due to the one high DWL
outlier at low sonic anemometer wind in panel (f). The two fast continuous measurement modes CSM2 and CSM3 yield
narrow scatterplots in panels (g) and (h), respectively, with low RMSE ($0.43\,\mathrm{ms^{-1}}$ and $0.34\,\mathrm{ms^{-1}}$), little bias ($-0.1\,\mathrm{ms^{-1}}$
and $0.12\,\mathrm{ms^{-1}}$), and low variation in terms of $R^2$ (0.98 and 0.99).

The DWL data availability at 90.3 m is close to 100 % for all configurations. Data availability with height depends mainly
on the elevation angle, i.e. the steeper the angle, the higher the amount of retrieved wind data at a certain height. For the
same elevation angles, the configuration with more pulses emitted per beam tends to achieve higher data availability for a
given height (compare DBS and CSM2, and 3Beam2 and 3Beam1, respectively). The 6Beam has a comparatively low data
availability in the vertical profile. However, it should also be mentioned here again that the data are not directly comparable
because the observation period and duration are different. The 6Beam observation period fell in November/December, which is
a period with different atmospheric conditions, especially more precipitation and the more frequent occurrence of low clouds
and fog, which can interfere with the DWL observations. All in all, the configurations seem to be able to properly monitor the
lowest 1 km above the ground and thus the atmospheric boundary layer.

Figure 8 shows scatterplots of the 10 min gust peaks from sonic anemometer versus the DWL retrieval for the eight con-
figurations. The performance of the configurations depends strongly on the time required per DWL circulation. The two slow
configurations CSM1 and 24Beam in panel (a) and (b) underestimate the gust peaks (biases of $-0.97\,\mathrm{ms^{-1}}$ and $-1.1\,\mathrm{ms^{-1}}$).
Here, the CSM1 yields a good coefficient of determination  (0.93) and could still be useful with an adequate bias correction,
while the 24Beam results appear too variable, especially for the highest gust peaks. DBS and 6Beam appear quite accurate
in panels (c) and (d) with lower RMSEs ($0.69\,\mathrm{ms^{-1}}$ and $0.86\,\mathrm{ms^{-1}}$). However, their observation periods coincide with weak
gust peaks, so their performance is not entirely clear. At least the highest gust peaks determined for the 6Beam are below
the intersection line, suggesting that more extreme gust peaks tend to be underestimated. Here, a bias correction or rescaling
could also provide useful results. Obviously, in too many cases, the 3Beam1 in panel (e) fails to detect the actual low gust
peaks recorded by the sonic anemometer. In contrast, though, the few actual high gust peaks are detected very well. The par-
allel measuring 3Beam2 in panel (f) provides only two significant overestimates, but is less capable of catching the highest



gust peaks, although it still gives reasonable results. Both 3Beam configurations thus provide worse performance values (e.g. RMSEs of $2.29\,\mathrm{ms^{-1}}$ and $1.36\,\mathrm{ms^{-1}}$). The fast CSM configurations are closest to the gust definition of a wind peak lasting at least 3 s since it takes 3.4 s to complete their measurement cycles. The two scatterplots in panels (g) and (h) show very high

agreement between the measured gust peaks from the DWL and sonic anemometer. Although the performance values (e.g. biases of $0.14\,\mathrm{ms^{-1}}$ and $-0.34\,\mathrm{ms^{-1}}$) are comparable to DBS and 6Beam, the measurements include gust peaks above $20\,\mathrm{ms^{-1}}$. Moreover, for the two elevation angles of $62°$ and $35.3°$ studied here, high gust peaks were observed whose points were also close to the intersection line. The linear fit for CSM2 is nearly perfect at the line of intersection, while the flatter CSM3 has a fit with a slightly lower slope. At the steep elevation angle, the observation cone at 90.3 m has a diameter of almost 100 m, and

at the lower elevation angle, a diameter of 255 m. The smaller the studied volume, the more likely one particular gust can be assumed to be detectable in the observation cone. In terms of RSME, the CSM2 provides the lower value ($0.77\,\mathrm{ms^{-1}}$ compared to $0.87\,\mathrm{ms^{-1}}$).

The availability of the wind data is generally lower than that of the mean horizontal 10 min wind, but again the same elevation angle dependence is evident, i.e., the higher the elevation angle of the configuration, the more data is available at given height.

Consideration of all these factors, combined with relatively good data availability in the vertical for an elevation angle of $62°$, leads to the decision to use the CSM2 for the later observation periods.

### 4.2 Extra-tropical cyclone Sabine

Storm Sabine was an extra-tropical cyclone with severe impacts throughout Europe. Gale-force winds led to the collapse of large sections of the transport network in Germany. The highest gust peak in Germany of about $49.1\,\mathrm{ms^{-1}}$ was measured at

Feldberg in the Black Forest (Haeseler et al., 2020). For the Falkenberg's sonic anemometer at 90.3 m, the highest gust peak was observed on 10 February 2020, at $29.3\,\mathrm{ms^{-1}}$, which was the highest value during the observation period of our study.

Figure 9 shows the observations during the three-day evolution of storm Sabine at 90.3 m in Falkenberg for both, a DWL operated in CSM2 and the sonic anemometer. It can be seen that the wind speed increases throughout the day on the 9th, reaching the overall highest values around noon on the 10th. During the following night, the wind intensity decreases, becoming

high again on the 11th and decaying afterwards (on the 12th which is not shown). The complete time series of the sonic anemometer is mostly correctly reproduced by the DWL in terms of the 10 min mean wind, the gust minima, and the gust peaks. There are three periods when the DWL underestimates the gust minimum and at the same time tends to underestimate the 10 min mean wind. Simultaneously, however, the gust peaks are adequately reproduced. Furthermore, the strongest gust is calculated to be $29.8\,\mathrm{ms^{-1}}$. It deviates by only $0.5\,\mathrm{ms^{-1}}$ from the sonic anemometer measurement, thus providing a convincing

result. For the other high gust peaks, in some cases larger deviations are registered, although these do not show any systematic under- or overestimation. In addition, the horizontal wind values of the individual cycles are shown, which cover the ranges of gust minimum to gust peak. As shown by the discrepancy of some DWL cycle winds and the returned DWL gust peaks/minima, the implemented outlier detection works and mostly unrealistic high or low values are filtered out before peaks and minima are determined.





To assess the performance of the retrieval in terms of vertical resolution of gusts, we compare our new retrieval to a classic retrieval exemplified for February 10, 2020 in Fig. 10. The wind barbs in panel (a) are from a classic retrieval with cycle-based MLE for prefiltered Doppler velocities, i.e. the SNR filter threshold is chosen at 18.2 dB according to Päschke et al. (2015). For each MLE, 66 % available Doppler velocities are required and for the calculation of the 10 min gust peak at least 50 % of the single circulations must have been processed. This retrieval represents a classic noise filtering approach but is not designed for

retrieving gusts. Panel (b) shows the result of our proposed retrieval. Both retrievals used the same measured Doppler velocities from a DWL operated in the CSM2 configuration. The new retrieval has a significantly higher data availability. The gust peaks indicated by the classic retrieval are very similarly covered by the new retrieval, which shows that the new retrieval eventually uses the same observations that the classic threshold filtering would leave. In panel (b), the additional obtained gust peaks fit coherently into the overall impression of the storm. This means that neither results that are also produced by a classic approach

are falsified, nor it is apparent that too many noisy values that are initially taken into account disturb the retrieval outcome.

This example is a satisfactory demonstration of the usefulness of the CSM2 configuration in combination with the new retrieval in terms of data availability of coherent gust peaks. Exactly such extreme events, are to be monitored precisely by the DWL. For this reason, and because of the statistics from the whole comparative test campaign, we set up a DWL in CSM2 throughout the summer of 2020.

**4.3    Summer 2020**

We extend the validation to a longer time period to look at a large sample of data. Figure 11 provides the comparative statistics for observations from summer 2020. The three months were relatively warm and dry for Brandenburg, while in addition weak winds from the North-Northeast prevailed frequently. This is reflected in the data availability for the comparisons, which is reduced by 13 % due to the shading effect of the tower on the sonic anemometer. The DWL, in contrast, conducts wind mea-

surements for almost the entire period. The vertical lines denote the sonic anemometer measurements in which the DWL does not process any winds and which are less than 1 % for both wind products. In particular, neither high mean winds nor strong gust peaks are missing in the processed data of the DWL. The comparison of the 10 min mean winds confirms that the CSM2 is suitable for deriving a mean wind at 90.3 m. Both the appropriate linear fit and the measures of spread, i.e. RMSE ($0.4\,\mathrm{ms^{-1}}$) and the coefficient of determination (0.98), emphasize the suitability of the retrieval and the used configuration for retrieving a

conventional DWL wind product. The comparison of the gust peaks provides high coincidences. The scatter is larger compared to the mean wind, as a small-scale process is more difficult to capture. As for the discussed test period, the CSM2 does not introduce a systematic error, and larger deviations are rare. Except for 18 cases, gust peaks are calculated for the situations in which the 10 min mean wind is processed. It can therefore be assumed that iterative filtering eliminates noise in a relatively similar way, regardless of whether the mean wind vector or the instantaneous wind value of an individual measurement cycle

is considered. Although it may happen that a high gust peak could generate Doppler velocities that are considered as noise in the derivation of the mean wind, it is precisely then that it is very practical to filter for both wind products independently. On the one hand, Doppler velocities of an individual gust peak that are significantly different to Doppler velocities belonging to the mean wind are negligible in the mean wind retrieval as these would be only few of the total amount of observations





within 10 min. On the other hand, that gust peak is recognized as such in the circulation based retrieval, provided it is clearly
visible in a single DWL cycle. Thus, the noise filtering seems to work effectively with respect to the requested wind product.

The scatterplot includes the uncertainty estimates for the horizontal winds. The standard deviation, shown with two vertical
bars for each point, should approximately cover the range by which the observation falls within 68 % probability for normally
distributed random variables. The estimation of the uncertainty depends on the choice of the effective DOF i.e. $n_{ef}$ from
Eq. (12). We set $n_{ef} = 12$ for the mean wind 10 min and $n_{ef} = 2$ for the wind of a circulation, which is then also represen-
tative for the gust peak. Except for some outliers, the uncertainties for both wind products emphasize the agreement between
DWL and sonic anemometer, and larger deviations between them are usually associated with larger uncertainties. The two
effective DOF used here are a result of tests with different $n_{ef}$. For that, we used all available results from observations in
the CSM2 configuration, i.e. also the measurements of the comparative test study. Figure 12 shows scatterplots for uncertainty
estimates against the difference between sonic anemometer and DWL wind, as well as an assessment from a probabilistic point
of view with rank histograms, namely for $n_{ef} = 12$ and $n_{ef} = 2$. In these rank histograms, the retrieval outcome is understood
as expectation and variance parameters of a Gaussian cumulative distribution function (CDF) which is evaluated at the sonic
anemometer observation. The histogram illustrates the frequencies of the different CDF values. An equally distributed rank
histogram indicates a calibrated forecast, i.e., in which the uncertainty parameters of the distribution are neither underestimated
nor overestimated. In panel (a) it can be seen that the 10 min mean wind is estimated to be very confident while also deviating
relatively little from the sonic anemometer observation. Nevertheless, it is also recognizable that tentatively more winds are
underestimated than overestimated. Such underestimated winds come with increased DWL uncertainty estimates, which be-
comes extremely noticeable in the case of one realization (cf. upper left corner). Differences and uncertainty estimates are of
the same order of magnitude, however. For the mean wind in panel (d) it is apparent that the sonic anemometer 10 min mean
wind is over-proportionally often higher than the expectation value. Nevertheless, setting the effective degrees of freedom
with $n_{ef} = 12$ results in an appropriate order of magnitude for effective independence. Higher values for $n_{ef}$ would reduce the
estimates for uncertainty and contribute to a slight flattening of the rank histogram, but also lead to a more frequent occurrence
of results around $CDF = 1$ (i.e., cases of underestimated winds with simultaneously too high estimated confidence). Vice
versa, a lower $n_{ef}$ would yield too high uncertainties, producing a higher peak in the rank histogram. The skewness cannot
be fixed with the modification of $n_{ef}$. Concerning the evaluation for gust peaks in panel (b), it is again noticeable that the
differences between the sonic anemometer and DWL are generally larger than for the mean winds. At the same time, however,
the estimate for the uncertainty also is larger. Further, it is apparent that gust peaks tend to be overestimated by the DWL.
Panel (e) confirms this impression, because there are more evaluations of the sonic anemometer observation on the left side of
the histogram. With $n_{ef} = 2$ we set a reasonably low value in order not to underestimate the uncertainty. There are not many
misunderstood outliers, i.e. there are not too many sonic anemometer gust observations that do not match the retrieval at all
and whose CDFs are close to 0 or 1. Since the consideration of extreme gust peaks is of particular relevance, panels (c) and (f)
show the assessment for gust peaks above $14\,\mathrm{ms}^{-1}$, i.e. gusts where to warn of in Germany. No significant difference to the
assessment of all gust peaks can be ascertained here. This confirms once again that strong gusts in our observation period do
not present special difficulties to the new retrieval.



# 5   Conclusions

Within the framework of the FESSTVaL measurement campaign, we investigate various configurations with regard to their ability to observe 10 min mean wind and wind gust peaks. For this purpose, a retrieval is developed that can flexibly quantify wind and associated uncertainty for different averaging time intervals. Our noise filtering is meshed in the retrieval and is based on the assumption that noise is distinguishable from real measurements and can be removed iteratively. The retrieval proves to be suitable to process the 10 min mean wind for all tested DWL configurations. Besides the mean wind, the retrieval

is used to process the wind of the single DWL cycles. The maxima of the single cycles within 10 minutes are considered to represent the wind gust peaks. Due to different settings, the tested configurations differs in the time required to complete all measurements of a respective cycle. A quick continuous scanning mode, the CSM2, proves to be successful for deriving gust peaks similar to those of a sonic anemometer at  90.3 m. This CSM2 provides 11 single radial Doppler wind measurements during one revolution of the DWL scan head, which is completed within 3.4 s, and from which the wind vector is derived.

Measurements with this configuration are performed during the passage of the extra-tropical storm Sabine in February 2020. The strongest gust peak in the whole observation period was measured here and accurately reproduced. Comparison of the new retrieval with a classic approach showed significantly higher vertical data availability for the new retrieval. Although comparative measurements from other heights are missing, the results of this storm day example provide a coherent overall picture of the vertical wind gust profiles. The other configurations require a longer time to complete a measurement cycle and

are therefore unsuitable for measuring wind gusts directly. However, it would be possible to scale the retrieved gusts to obtain values that are more comparable to the 3 s sonic anemometer results. In particular, the scaling method of Suomi et al. (2017) could be applied.

During the summer of 2020, we tested the CSM2 for full three months. For both mean winds and gust peaks, we are able to cover almost the entire observation period for which usable sonic anemometer observations exist. Overall, the DWL and

sonic measurements agreed with low RMSE (0.4 ms$^{-1}$ for 10 mean wind and 0.8 ms$^{-1}$ for gust peaks, respectively) and small biases ($-0.24$ ms$^{-1}$ and 0.32 ms$^{-1}$) and in addition there are also no cases of strong gusts that the DWL retrieval has not identified. Finally, the estimated uncertainty of the retrieval is evaluated. The uncertainty estimates for mean wind and gust peaks are in the order of magnitude of absolute error with respect to the sonic anemometer. The mean wind is somewhat too often underestimated by the DWL while the gust peaks are rather too often detected higher than the sonic anemometer. Apart

from this asymmetry, these results are nevertheless satisfactory, because it also shows that the DWL distribution did not too often describe situations that do not match the sonic anemometer observation.

The uncertainty was correctly represented, but the use of an effective DOF is necessary. We have used different DOFs for requested winds, i.e. whether it was cycle-based or within 10 minutes, but there is still room for improvement. Our aim was to provide a reliable estimate and its tuning is beyond the scope of this study. In particular, separate DOFs could also be

appropriate for different weather situations, as well as for the different configurations, of which we examined only CSM2. We show how useful the CSM2 could be, if operated at an elevation angle of 62°. Using an elevation angle 35.3° gives results of similar quality. We have not systematically answered how to choose the optimal angle, which could be investigated further.



The general advantage of the suggested fast CSM lies in the fact that it completes one measurement cycle within 3.4 s. To our knowledge, there is no comparable DWL scan configuration that performs a similar number of radial velocity measurements

in such a short cycle.

The newly available FESSTVaL data set from summer 2021 offers further opportunities for detailed case studies and comparative studies involving several DWLs and airborne in-situ measurements. The airborne measurements provide a reference for the quality of retrieval in higher layers. There are parallel DWL measurements in the same quick CSM configuration, but at different locations, so that the spatial evolution of gust structures can be analyzed. In a study Steinheuer and Friederichs (2020)

show that gust profiles can be derived from reanalysis data. This method can still be tested at various locations, which is now also possible with the means of DWLs. We hope that our retrieval lays the foundation for expanding the monitoring network for high-frequency wind measurements with DWLs for weather research and applications.

**Appendix**

The expected value of $\hat{\mathbf{v}}_0$ holds

$$
\begin{aligned}
\quad E[\hat{\mathbf{v}}_0] &= E\left[(A^T A)^{-1} A^T \mathbf{d}\right] & (13) \\
&= E\left[(A^T A)^{-1} \sum_{i=1}^{n} \mathbf{a}_i d_i\right] & (14) \\
&= E\left[(A^T A)^{-1} \sum_{i=1}^{n} \mathbf{a}_i (\mathbf{a}_i^T \mathbf{v}_i + \epsilon_i)\right] & (15) \\
&= (A^T A)^{-1} \sum_{i=1}^{n} \mathbf{a}_i (\mathbf{a}_i^T \mathbf{v}_0 + 0) & (16) \\
&= (A^T A)^{-1} A^T A \mathbf{v}_0 & (17) \\
\quad &= \mathbf{v}_0. & (18)
\end{aligned}
$$

Equations (13) - (15) are obtained by inserting definitions. Then the expectation is applied on the $\mathbf{v}_i$ and $\epsilon_i$ and the matrices cancel out.





The variance of $\hat{\mathbf{v}}_0$ holds

$$
\begin{aligned}
Cov\left[\hat{\mathbf{v}}_0\right] \; &= \; E\left[\left((A^TA)^{-1}A^T\mathbf{d} - \hat{\mathbf{v}}_0\right)\right. \\
&\qquad \left.\left((A^TA)^{-1}A^T\mathbf{d} - \hat{\mathbf{v}}_0\right)^T\right] 
\end{aligned}
\tag{19}
$$

$$
\begin{aligned}
&= \; E\left[\left((A^TA)^{-1}\sum_{i=1}^{n}\mathbf{a}_i\left(\mathbf{a}_i^T\mathbf{v}_i + \epsilon_i\right) - \hat{\mathbf{v}}_0\right)\right. \\
&\qquad \left.\left((A^TA)^{-1}\sum_{i=1}^{n}\mathbf{a}_i\left(\mathbf{a}_i^T\mathbf{v}_i + \epsilon_i\right) - \hat{\mathbf{v}}_0\right)^T\right]
\end{aligned}
\tag{20}
$$

$$
\begin{aligned}
&= \; E\left[\left((A^TA)^{-1}\sum_{i=1}^{n}\mathbf{a}_i\left(\mathbf{a}_i^T\mathbf{v}_i + \epsilon_i - \mathbf{a}_i^T\mathbf{v}_0\right)\right)\right. \\
&\qquad \left.\left((A^TA)^{-1}\sum_{i=1}^{n}\mathbf{a}_i\left(\mathbf{a}_i^T\mathbf{v}_i + \epsilon_i - \mathbf{a}_i^T\mathbf{v}_0\right)\right)^T\right]
\end{aligned}
\tag{21}
$$

$$
\begin{aligned}
&= \; \sum_{i=1}^{n}E\left[(A^TA)^{-1}\left(\mathbf{a}_i\left(\mathbf{a}_i^T\left(\mathbf{v}_i - \mathbf{v}_0\right) + \epsilon_i\right)\right)\right. \\
&\qquad \left.\left((A^TA)^{-1}\mathbf{a}_i\left(\mathbf{a}_i^T\left(\mathbf{v}_i - \mathbf{v}_0\right) + \epsilon_i\right)\right)^T\right]
\end{aligned}
\tag{22}
$$

$$
= \; \sum_{i=1}^{n}(A^TA)^{-1}\mathbf{a}_i\left(\mathbf{a}_i^T\Sigma\mathbf{a}_i + \sigma_\epsilon^2\right)\mathbf{a}_i^T(A^TA)^{-1}
\tag{23}
$$

$$
= \; \sum_{i=1}^{n}(A^TA)^{-1}\mathbf{a}_i\sigma_d^2\mathbf{a}_i^T(A^TA)^{-1}
\tag{24}
$$

$$
= \; (A^TA)^{-1}(A^TA)\sigma^2(A^TA)^{-1}
\tag{25}
$$

$$
= \; (A^TA)^{-1}\sigma^2.
\tag{26}
$$

Equation (21) arises by supplementing $\mathbf{v}_0$ with $(A^TA)^{-1}A^TA$ and rearranging. Then in Eq. (24) the expectation value calculation is applied, exploiting that observation errors $\epsilon_i$ are uncorrelated to each other and to the individual deviations from the mean wind $(\mathbf{v}_i - \mathbf{v}_0)$.

*Code availability.* The code is available at: https://doi.org/10.5281/zenodo.5780949 (Steinheuer et al., 2021a).

*Data availability.* The data is available at: https://doi.org/10.25592/uhhfdm.9758 (Steinheuer et al., 2021b).

*Author contributions.* UL and FB, together with others, planned the FESSTVaL campaign. FB and CD were responsible for logistics on the site and performed the measurements. JS and CD developed the retrieval methodology with idea contributions from all others. JS coded the



retrieval. JS, UL, and SF planned and structured the paper. JS and PF developed the uncertainty methodology of the retrieval. JS drafted the manuscript. JS, CD, FB, UL, PF, and SF reviewed it iteratively.

*Competing interests.* The authors declare that they have no conflict of interest.

*Acknowledgements.* Especially we would like to thank Ronny Leinweber (DWD, MOL-RAO) for the installation and maintenance of the Doppler Wind lidars. We are grateful to Fred Meier (Technical University Berlin, Institute for Ecology) for providing us with a Doppler Wind lidar in autumn 2019. Jan Schween (University of Cologne, Institute for Geophysics and Meteorology) always provided helpful assistance with all DWL-related questions. We thank Markus Kayer (DWD, MOL-RAO) and Ronny Leinweber for providing the Level 1 data. There
were useful ideas from Eileen Päschke (DWD, MOL-RAO) and valuable discussions with all mentioned. Also, this work has profited from the scientific exchange within the EU COST Action PROBE (CA18235).

*Financial support.* This work has been conducted in the framework of the Hans Ertel Centre for Weather Research funded by
the German Federal Ministry for Transportation and Digital Infrastructure (grant no. BMVI/DWD 4818DWDP5A).



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


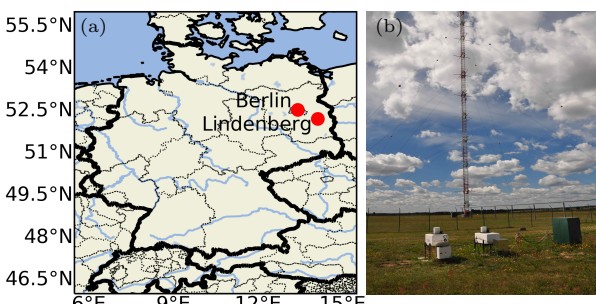

**Figure 1.** Measurement site. Panel (a): The MOL-RAO is situated in the northeastern part of Germany, approximately 65 km southeast of Berlin. Panel (b): Two Doppler wind lidars in front of Falkenberg meteorological tower (July 27, 2020, Author's photo); Falkenberg is approximately 5 km south of Lindenberg.





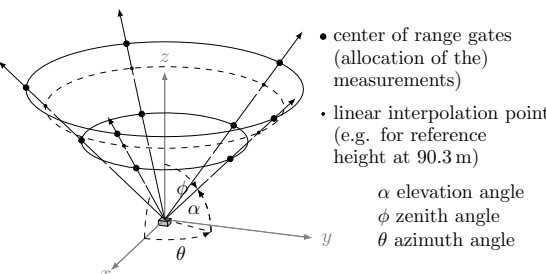

**Figure 2.** The DWL observation principle is shown here with five beams per cycle. Each beam consists of several thousands of laser pulses and the backscattered signal is affiliated to a discrete series of range gates depending on the length of a single laser pulse and the travelling time. The resulting Doppler velocities are assigned to center of range gates. In order to obtain comparable results at intermediate heights, a linear interpolation between the two neighbouring measurements along each beam is performed.



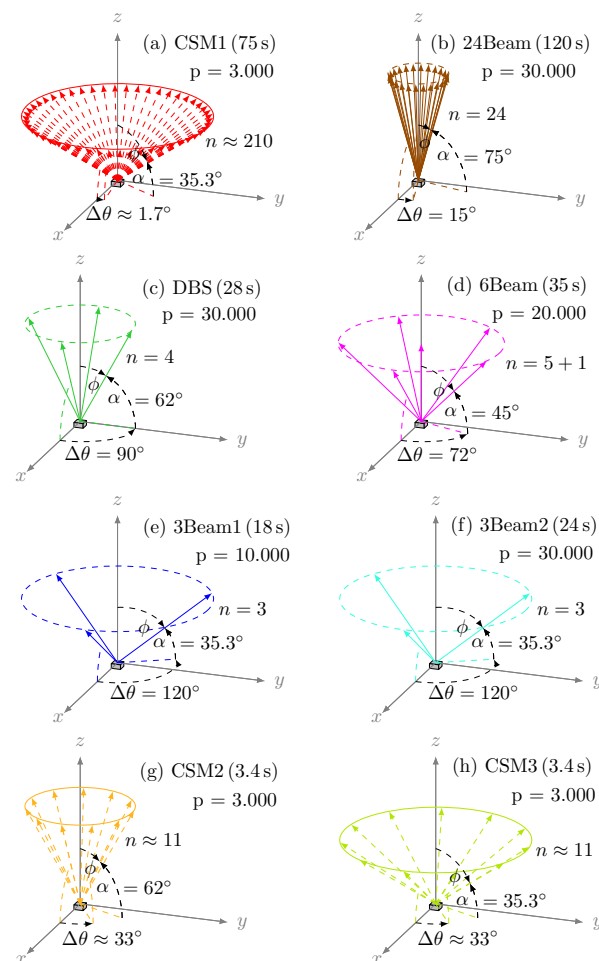

**Figure 3.** The different tested DWL configurations with corresponding averaged cycle time (in parentheses), the total number of averaged pulses per measured Doppler velocity (p), their elevation angle ($\alpha$) and azimuth step-angle ($\Delta\theta$), and the number of beams per cycle (n, for the continuous modes this is an approximated value).



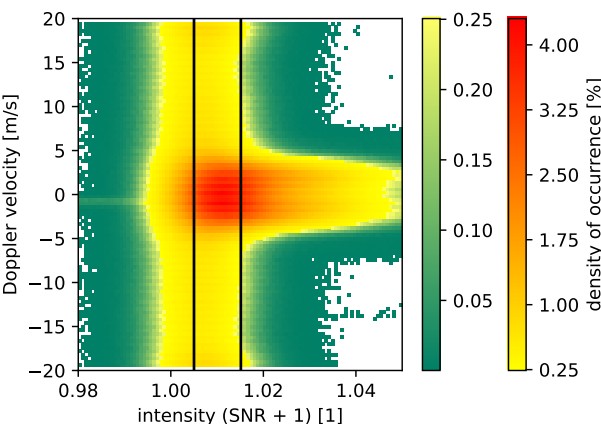

**Figure 4.** Intensities (SNR + 1) vs. Doppler velocities on September 2, 2019 for all center-of-range gates during a 24 h observation period. The DWL is operated in CSM2 with $62°$ zenith angle and it produced 25 million measurements on that day. The area is divided into $100 \times 100$ bins and the colors indicate the density of occurrence. The left vertical line corresponds to an SNR value of $-23\,\mathrm{dB}$ and the right to $-18.2\,\mathrm{dB}$.





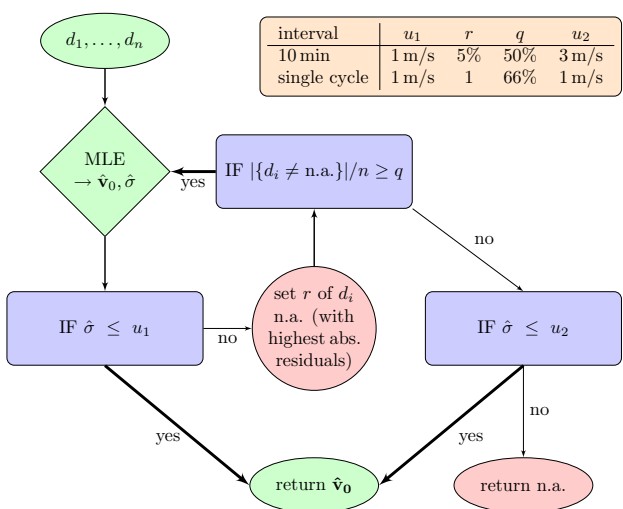

**Figure 5.** Schematic flowchart of the DWL retrieval with the main steps to determine the wind vector estimate $\hat{\mathbf{v}}_0$. All measurements $d_1, \ldots, d_n$ from a given height and in a given time interval pass through the iteration loop. If-statements (blue) use thresholds ($u_1$, $u_2$, and $q$) to decide whether to set n.a. values (red) or pass wind-fit data (green). The thresholds depend on whether the time interval is $10\,\mathrm{min}$ or consists only of the measurements of a single DWL cycle (see orange box).



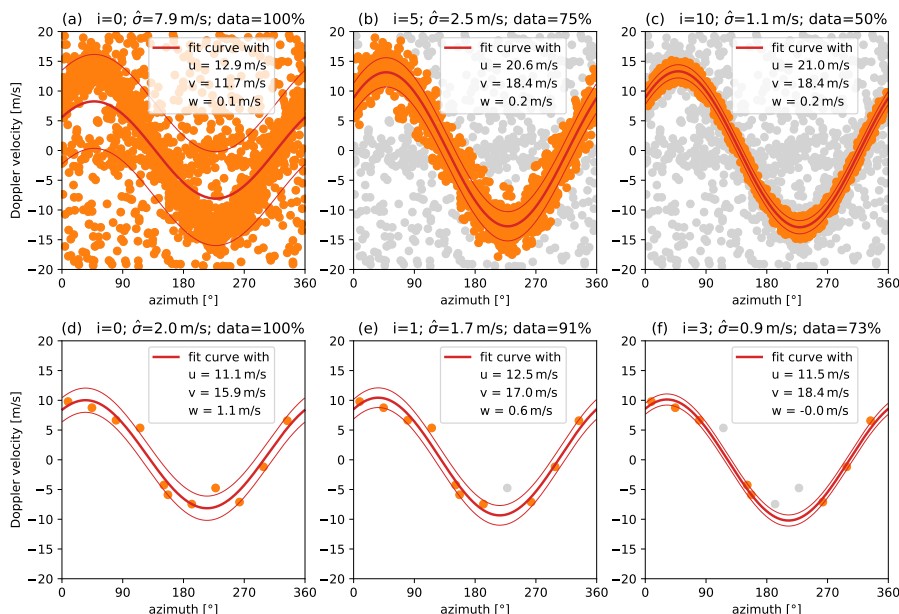

**Figure 6.** Different steps (i) of the retrieval for 10 min mean wind (top) and for the wind of a single DWL cycle (bottom). The sinusoidal projection (fit curve with) of the fitted wind vector (u,v, and w) is shown in thick red, with a standard deviation tube around it ($\pm\hat{\sigma}$, red). Used observations for the displayed wind vector fit are orange and omitted observations are grey. The images (c) and (f) display the fits that are finally returned. The examples are from a DWL operated in CSM2 on February 10, 2020, and from measurements at 808 m (top) and 225 m (bottom), respectively.

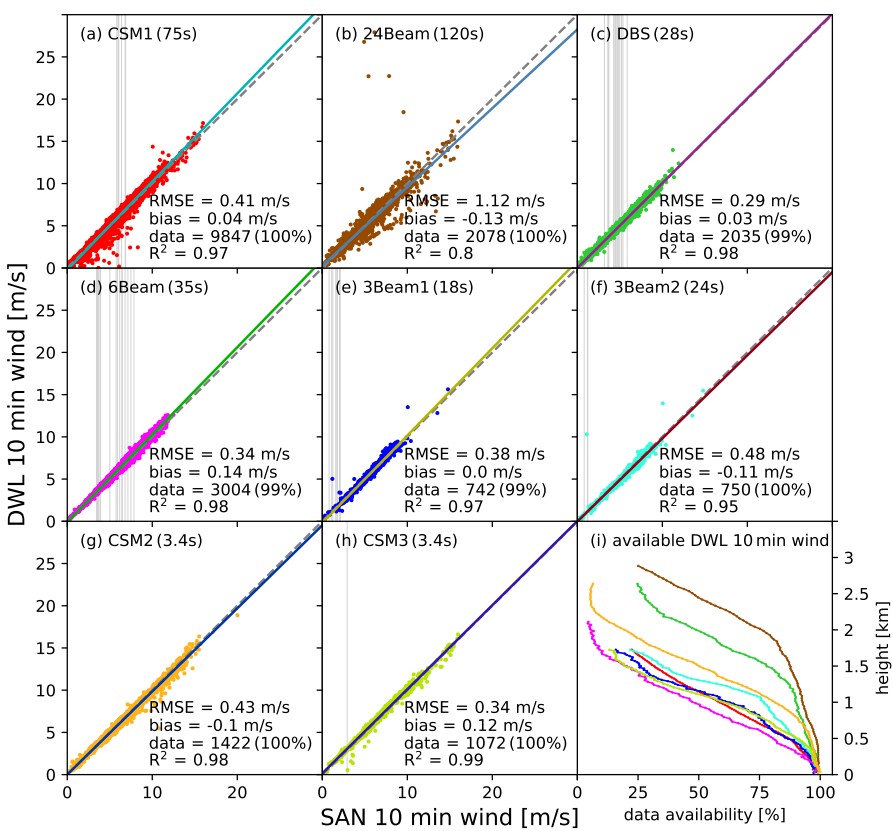

**Figure 7.** Scatterplots of 10 min mean horizontal wind from sonic anemometer (SAN) versus DWL for the eight different tested DWL configurations at 90.3 m. Colors and letters (a)-(h) correspond to the configurations shown in Fig. 3 with the measurement configuration schedule given in Table 1. For each panel, coloured linear fit line, the root-mean-squared error (RSME), the bias, the involved data, and the coefficient of determination ($R^2$) are given. The parameter data indicates in parentheses the fraction of situations where the DWL retrieval returned valid wind values. Grey vertical lines indicate sonic anemometer measurements with missing corresponding DWL results. Panel (i) shows the DWL data availability against height with colors per configuration as in (a)-(h).



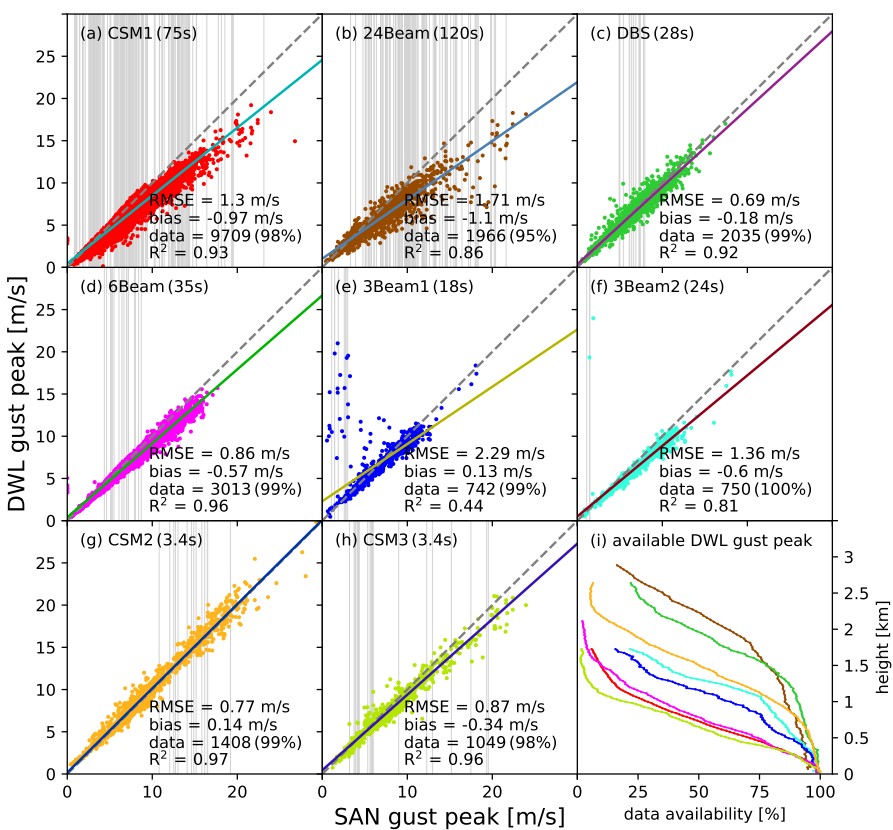

**Figure 8.** Scatterplots of the sonic anemometer (SAN) gust peak (3 s in 10 min) versus the DWL gust peak (gust duration as indicated per panel in 10 min) for the eight different tested DWL configurations at 90.3 m. The further explanations are the same as in Fig. 7. Panel (i) shows the data availability against height with colors per configuration as in (a) - (h).

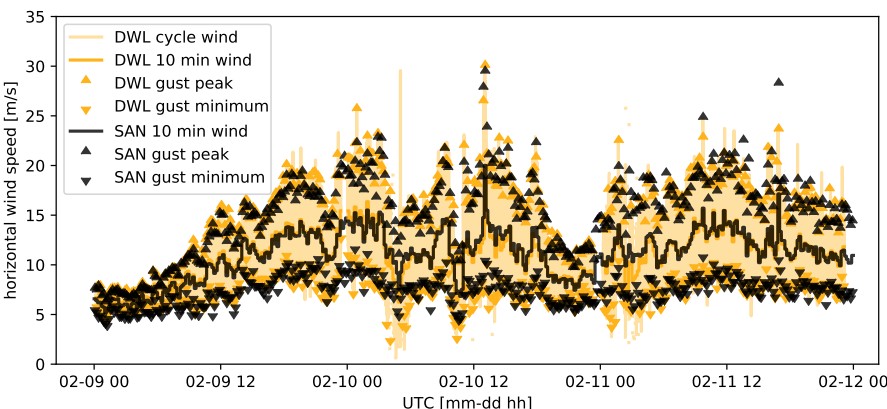

**Figure 9.** Time series of wind speeds during extra-tropical storm Sabine on February 9 – 11, 2020. Both, DWL operated in CSM2 (yellow) and sonic anemometer (black) winds are shown. The triangles indicate the 10 min gust peaks and minima, the thick solid lines indicate the 10 min mean horizontal wind, and the light yellow line shows the results for all DWL cycles. Note that due to outlier filtering, not all cycle maxima and minima match the peak and minimum gust values.

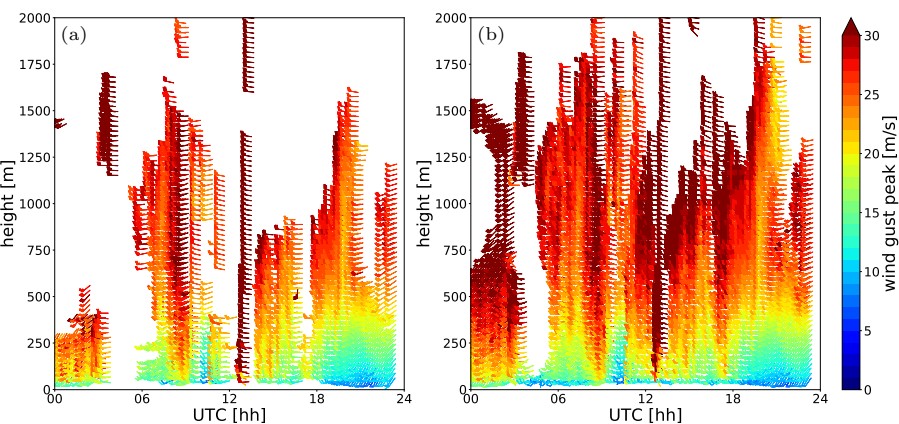

**Figure 10.** Color-coded wind barbs for gust peaks during extra-tropical cyclone Sabine on February 10, 2020 from DWL operated in CSM2. Panel (a): Results for a classic retrieval with SNR filtering at $-18.2$ dB and least squares fit. Panel (b): Results without SNR filtering and iteratively improved least squares fit as developed in this study.

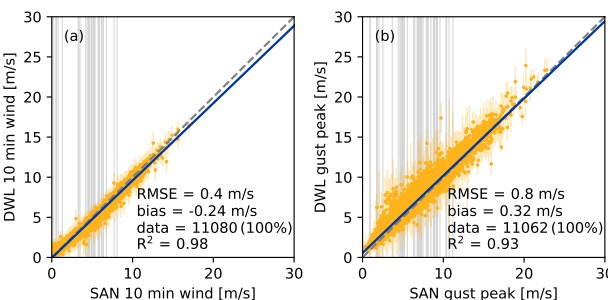

**Figure 11.** Scatterplots of sonic anemometer vs. DWL wind retrieval during the period June 1, 2020 to August 31, 2020 at 90.3 m. Panel (a): Scatterplot of 10 min mean horizontal wind from sonic anemometer versus DWL operated in CSM2. Panel (b): Scatterplot of the sonic anemometer gust peak (3 s in 10 min) versus the DWL gust peaks (3.4 s) operated in CSM2. The diagnosis numbers are explained in Fig. 7. The estimated DWL standard deviation of the horizontal wind or gust peak is shown with vertical bars derived from the estimated covariance matrix with $n_{ef} = 12$ (a) and $n_{ef} = 2$ (b), respectively.

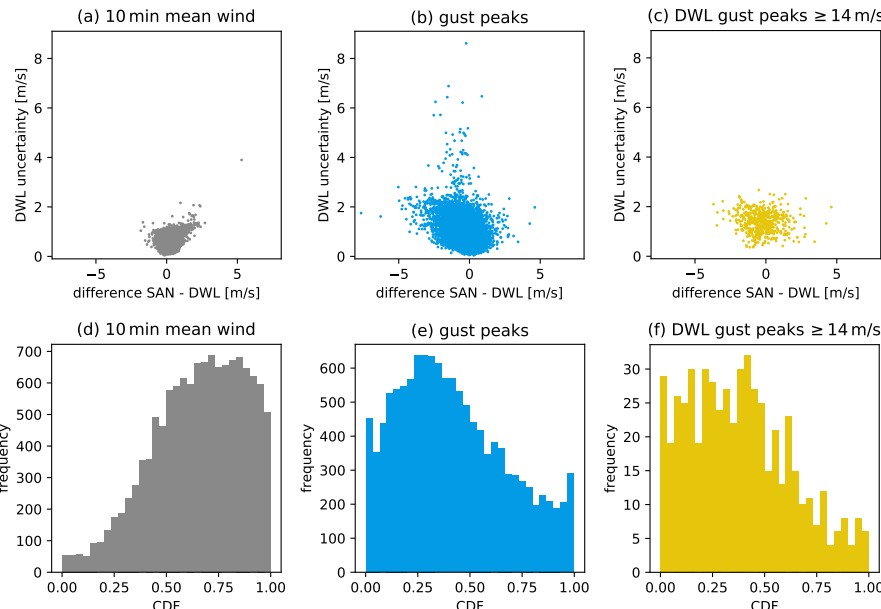

**Figure 12.** Examination of uncertainty. The top panels show comparisons for the differences of the sonic anemometer and DWL wind values against the estimated DWL uncertainty. Panel (a) displays the results for the 10 min mean horizontal wind, panel (b) for the gust peaks, and panel (c) for only cases where the DWL gust peaks exceed $14\,\mathrm{ms}^{-1}$. The bottom panels show the rank histograms for the retrieved DWL wind and its corresponding uncertainty, where panel (d) addresses the 10 min mean horizontal wind, panel (e) gust peaks, and panel (f) the gust peaks exceeding $14\,\mathrm{ms}^{-1}$. Each histogram shows the frequency of the Gaussian wind cumulative distribution function values, evaluated at the sonic anemometer observations. A perfect model would show equally distributed frequencies.





**Table 1.** Instrument parameters of the three HALO Photonics StreamLine DWL systems.

|  | DWL 78 & DWL 177 | DWL 143 |
| --- | --- | --- |
| Instrument type | StreamLine | StreamLine XR |
| Wavelength | $1.5\,\mu$m | $1.5\,\mu$m |
| Pulse width | 180 ns | 352 ns |
| Range gate length | 30 m | 30 m |
| Pulse repetition frequency | 10 kHz | 10 kHz |
| Resolution of Doppler velocity | $\pm0.038\,\text{ms}^{-1}$ | $\pm0.076\,\text{ms}^{-1}$ |
| Telescope focus | 2000 m | 6535 m |
| Sampling frequency | 50 MHz | 100 MHz |
| Nyquist velocity | $19.4\,\text{ms}^{-1}$ | $38.8\,\text{ms}^{-1}$ |
| Number of FFT points | 1024 | 1024 |



**Table 2.** Configuration time schedule for the used DWLs.

|  | CSM1 | 24Beam | DBS | 6Beam | 3Beam1 | 3Beam2 | CSM2 | CSM3 | days |
|---|---|---|---|---|---|---|---|---|---|
| 15/8 – 21/8/19 | DWL 78 |  |  |  |  |  |  |  | 7 |
| 22/8 – 26/8/19 | DWL 78 |  | DWL 177 |  |  |  |  |  | 5 |
| 29/8 – 5/9/19 | DWL 78 | DWL 143 |  |  |  |  | DWL 177 |  | 8 |
| 7/9 – 17/9/19 | DWL 78 |  | DWL 177 |  |  |  |  |  | 11 |
| 18/9 – 22/9/19 | DWL 78 |  |  |  |  |  |  |  | 5 |
| 23/9 – 30/9/19 | DWL 78 | DWL 143 |  |  |  |  |  | DWL 177 | 8 |
| 1/10 – 7/10/19 | DWL 78 |  |  |  | DWL 177 | DWL 143 |  |  | 7 |
| 19/11 – 12/12/19 | DWL 78 |  |  | DWL 177 |  |  |  |  | 24 |
| 9/2 – 11/2/20 (Sabine) |  |  |  |  |  |  | DWL 177 |  | 3 |
| days test campaign | 75 | 16 | 16 | 24 | 7 | 7 | 11 | 8 |  |
| 1/6 – 31/8/20 |  |  |  |  |  |  | DWL 177 |  | 92 |
| days FESST@MOL |  |  |  |  |  |  | 92 |  |  |