# Peer review of "A new scanning scheme and flexible retrieval for mean winds and gusts from Doppler lidar measurements"

_Atmospheric Measurement Techniques, 2021_

## Author Response (AR2)

Anonymous Referee #1, 17 Feb 2022

We thank the reviewer for her helpful comments. Your suggestions are greatly appreciated and lead to improvement of the article. In the following, we respond (normal font) to your comments (**bold**) and provide the manuscript changes (*italic*).

**The manuscript propose a novel algorithm for the retrieval of the mean wind and wind gusts from wind-lidars. It is an outstanding paper, clearly written in a good english. The proposed retrieval algorithm for wind speed and wind gusts has the potential to be applied and replace the existing systems in future wind-lidar systems. The proposed algorithm include an uncertainty estimation of the wind speed.**

Thank you for the very positive feedback! We are pleased to hear that our retrieval can be valuable to the lidar community in the future. We will continue to work with retrieval and hope to show more results.

**Some minor remarks:**

**Line 209. Use "elevation" or "zenith" angle throughout the manuscript – not a mix.**

Zenith is now mostly omitted throughout the manuscript and replaced by elevation (as well as 'phi' is therby replaced by 'alpha'). Further, to avoid double-use in Section 3.4, 'gamma' is used instead of 'alpha'.

**Line 344-346: the sentence starting The outliers…..is not clear. How close is the lower range gate to the DWL?????? And why is this a problem.**

Changed: *For the 24beam in panel (b), some DWL outliers can be recognized, which can be explained by the relatively steep elevation angle. The outliers result from the fact that the linear interpolation of the Doppler velocities fails at 90.3 m because the involved Doppler velocities of the lowest range gate centers are too close to the DWL. Close to the DWL, transmitter and receiver field of view do not completely overlap. Therefore, the Doppler velocities originating from the lowest range gates should be discarded and those of the following ones are at least noisier. The amount of full overlap is instrument dependent, and the obvious outliers show that the Doppler velocities cannot always be considered reliable at 75 m radial distance from the DWL, i.e., at 72 m a.g.l. for 75° elevation, which corresponds to the distance to the third center of the range gate. In fact, a comparison with the results of range gates centered at 101 m a.g.l. (fourth center of range gates) would give a better result (not shown).*

**Figure 9. Explain the acronym SAN. The acronym should be explained in the main text or on every figure where it is used.**

SAN stands for sonic anemometer. We now introduce the acronym in every figure captions if used.

**Line 410. Should be -18.2 dB**

Yes, thanks!

**Conclusion: I suggest to include a small discussion on the use of CSM2 for turbulence measurements?**

Thank you for this suggestion. The turbulence characteristics are currently under investigation in the framework of FESSTVaL. We add the following in the conclusion: *Here, the high-resolution*

*time series of the wind vector generated with the retrieval offers the potential to study turbulence in detail. Thereby, it has to be shown whether the derived vertical wind is of comparable quality as measurements of a vertically pointing DWL.*

**Anonymous Referee #2, 07 Apr 2022**

We thank the reviewer for her helpful comments. Your suggestions are greatly appreciated and lead to improvement of the article. In the following, we respond (normal font) to your comments (**bold**) and provide the manuscript changes (*italic*).

**Development of a new method to retreive gusts from Doppler wind lidar measurements is a useful contribution to the field. The approach is interesting and fairly well outlined by the authors, with a few minor issues resulting in a recommendation for minor revision.**

**Generally, my only comments are on the presentation of the method in text. There are some inconsistencies in the writing and clarity throughout the paper. For example, the grammar and style in the initial sections is sometimes bumpy, but very polished in the sections outlining the retrieval method itself. On the other hand the content in the initial sections is well thought out in guiding the reader while the sections about the retrieval method assume a lot about reader's background knowledge in the methods. Perhaps this isn't a bad assumption, but it is different in tone and style than other sections. Overall I think these sections could be more integrated regarding style and voice to improve flow and readability.**

**In terms of the content itself, I am very excited about this method. I did find myself wanting to see more observations processed through the retrieval; but I do understand space limitations and focus may preclude this. If it is possible to show more observations without distracting from the narrative, I think this would be good to futher illustrate the utility of the method and point out the ability of the retreival to estimate gusts and mean wind. In the various parts of the paper that mention assumptions required for Doppler lidar observation, I felt it got a little lost actually how some of those assumptions are still honored for this gust retrieval method while still allowing the gust data to be retained.**

Thank you for the positive and constructive review. We have revised the language in the first section. Please refer to the manuscript with track changes to review the revised text. In the Retrieval-section we want to describe the assumptions very correctly and to formulate them mathematically consistently. Although this is stylistically more straightforward, it is admittedly different in style. We consider it necessary to give high weight to the retrieval along with the underlying assumptions.

We are glad to read that we arouse interest in further results. We are currently working on analyzing more case studies and presenting more research, but had to limit ourselves in this paper at a certain point. Other articles using data from this retrieval will be available, e.g., Weide Luiz and Fiedler, which is currently under review (see https://doi.org/10.5194/wes-2022-26) and examines the spatio-temporal characteristics of nocturnal low-level jets.

**Specific comments:**

**47 - phrasing unclear: "only in certain of these air parcels."**

Changed to: *Accordingly, small-scale wind variations may be noticeable only in certain regions of the sampled air volume and not all determined Doppler velocities may be affected the same way.*

**48 - not sure what it means when you said large spatial extent influence different measurements and thus be more detectable. Consider rephrasing this to be more clear.**

Changed to: *For the strongest gust peaks, we assume that they also occur over a larger area by realizing that the air parcels with increased velocities travel a longer distance in a given time interval. Thus, we assume that strong gusts influence the measured Doppler velocities to a sufficient extent over the whole sampling volume.*

**52 - Do you mean "The DWL they considered operated for two days…"? Using 'investigated' and 'scheduled' here is a little confusing.**

Changed as suggested.

**53 - Wind gusts, then gust peaks are derived? Should "wind gusts" be "Winds" from which the "gust peaks" are obtained? This is a bit unclear here.**

Revised to: *Wind vectors are derived from each set of five measurements, and gust peaks are obtained from them.*

**54-55 - Phrases from "The approach includes…" to "… meteorological tower": This is pretty unclear to the reader as written here.**

Rewritten to: *The approach includes a scaling method for the detected 3.8 s lasting gusts to infer 3 s lasting gusts. This way, the results agree well with 3 s lasting gust peaks as measured by a nearby sonic anemometer on a meteorological tower.*

**73 - Was this meant to be practical?**

This parts now it reads: *All results are derived from the new retrieval, which, in addition to calculating the gust peaks, is also used to determine the 10 min mean wind, since a practical configuration must also correctly capture the mean wind.*

**87 - Which DWL?**

Changed: *In 2020, one of these DWLs was in operation when extra-tropical cyclone Sabine passed in February, and during the three summer months.*

**168-172 - You write it can be seen that tripling the transmission rate does not increase the total cycle time. This isn't immediately clear.**

The pulses per beam are tripled. The sentence is now changed to: *It can be seen that tripling the transmission pulse rate from 10.000 to 30.000 pulses per ray does not increase the total cycle time significantly, or expressed differently, no time resolution close to a 3 s gust duration can be achieved with the devices when they are operating in the step-stare modes. In this mode most time is spent to accelerate the scan head, move it to the new position and slow down again to zero rotation speed.*

**397 - missing a space between 'both' and 'a' and the comma there is unneeded**

Thanks, it is adjusted accordingly.

**401 - what is 'mostly correctly' in terms of reproduction of a time series? Not very precise.**

Agreed. We have changed this introductory sentence and wrote: 'convincingly reproduced'. The following sentences then specify what is not 'correct'.

**402 - "Underestimates the gust minimum:" this language is difficult. Already, a 'gust minimum' is confusing, since gusts are synonymous with maxima. Is this really the best term? Now adding the proximal language of under or over estimation to this is very confusing. I quickly polled some colleagues about over- or under- estimating a minimum and where they thought those estimates would fall relative to the minimum. There was not agreement. I strongly recommend revisiting this language to avoid confusion. This is especially true in this case since the accompanying figure is a bit crowded, so contextual information wouldn't necessarily help readers deduce the answer.**

Good advice. We now use 'minimum wind' instead of 'gust minimum'.

**412 - Has MLE been defined?**

*Yes. It is defined in the Retrieval Section: MLE = maximum likelihood estimation*

**414 - Is the classic method retrieving gusts or not? The text and the figure caption both are a bit conflicting.**

The classic method was adapted and modified so that gusts could be retrieved. We changed parts of the paragraph and the figure caption to avoid confusion in the revised manuscript. The text now reads: *A classic retrieval is not designed to derive wind gusts, but usually to determine a mean wind, so the filtering can eliminate more measurements. Here, by classic retrieval, we mean classic threshold filtering followed by MLE, which determines the wind vector from the remaining measurements of each DWL cycle. Thus, similar to the new approach, we obtain wind vectors from which wind gusts can be derived. Hence, the calculation is not iterated and all remaining observations are used. The wind gusts in panel (a) are from this classic retrieval with the cycle-based MLE for prefiltered Doppler velocities at an SNR threshold at -18.2 dB according to Paeschke et al. (2015). For each MLE, 66 % available Doppler velocities are required, and for the calculation of the 10 min gust peak, at least 50 % of the individual cycles must have been processed (valid for both approaches). This procedure is a classic noise filtering, but with a calculation based on very few observations.*

The figure caption changed to: *Panel (a): Results for a retrieval with classic SNR filtering at -18.2 dB and MLE. Panel (b): Results without SNR filtering and iteratively improved MLE as developed in this study. In both approaches, the gust peak per 10\ min is only given if at least 50 % of the DWL cycles obtain valid values.*

**420 - Last sentence of this paragraph is very confusing. Not sure what it says.**

Changed to: *This means that the new retrieval does not distort results that are also produced by rigorous filtering. On the other hand, it is also not observed that the inclusion of too many observations that are potentially noisy disturbs the retrieval result.*

**444 - What is circulation based referencing?**

We distinguish between the 10 min retrieval and the cycle-based retrieval with only 11 single radial velocity measurements. To avoid confusion, we omit the word 'circulation' and use only 'cycle' throughout the text. The sentence changed to: *On the other hand, that gust peak is recognized as such in the single cycle based retrieval, provided it is clearly visible in the few measurements within one single DWL cycle.*

**476 - What is gusts where to warn in Germany?**

The DWD warns if gusts at 14 m/s or higher are predicted. We changed the sentence to: *Since the consideration of extreme gust peaks is of particular relevance, panels (c) and (f) show the assessment for gusts above 14 m/s. This is the threshold value for the forecast at which warnings of gusts must be issued in Germany.*

**Fig 4 - The double colorbar is odd. Also green-to-red is not very accessible to color-deficient readers. Explore more accessible color tables. This is also true for some of your other figures' line color combinations.**

Agreed, we have changed the color use in our figures 3, 4, 7, 8, 9, 10, and 11.

**Figure 9 - There are a lot of lines and markers here making it hard to really see what's going on. The main points are sufficiently shown I suppose… but is this the best visualization? In many instances, I couldn't tell if there even were other markers behind black markers.**

Yes, the usually very good agreement between the data means that markers and lines can completely overlap in places. We have now changed the colors and line widths for clarity. It does not fix every aspect raised for this Figure. We add in the text for the interpretation of Figure 9: *Very good agreement between the data means that markers and lines completely overlap.*

**Fig 10 - Colorbar could be improved. Research has shown rainbow bars to be deficient.**

Agreed, the colorbar has been adjusted.